# Modular and cloud-based bioinformatics pipelines for high-confidence biomarker detection in cancer immunotherapy clinical trials

Cu Nguyen[1], Trinh Nguyen[1], Gloria Trivitt[2], Brian Capaldo[1], Chunhua Yan[1], Qingrong Chen[1], Nicholas Renzette[2]*, Umit Topaloglu[2], Daoud Meerzaman[1]

**1** The Computational Genomics and Bioinformatics Branch, Center for Biomedical Informatics and Information Technology, National Cancer Institute, Rockville, Maryland, United States of America, **2** The Clinical and Translational Research Informatics Branch, Center for Biomedical Informatics and Information Technology, National Cancer Institute, Rockville, Maryland, United States of America

⊙ These authors contributed equally to this work.
* nick.renzette@nih.gov

## Abstract

### Background

The Cancer Immune Monitoring and Analysis Centers – Cancer Immunologic Data Center (CIMAC-CIDC) network aims to improve cancer immunotherapy by providing harmonized molecular assays and standardized bioinformatics analysis.

### Results

In response to evolving bioinformatics standards and the migration of the CIDC to the National Cancer Institute (NCI), we undertook the enhancement of the CIDC's extant whole exome sequencing (WES) and RNA sequencing (RNA-Seq) pipelines. Leveraging open-source tools and cloud-based technologies, we implemented modular workflows using Snakemake and Docker for efficient deployment on the Google Cloud Platform (GCP). Benchmarking analyses demonstrate improved reproducibility, precision, and recall across validated truth sets for variant calling, transcript quantification, and fusion detection.

### Conclusion

This work establishes a scalable framework for harmonized multi-omic analyses, ensuring the continuity and reliability of bioinformatics workflows in multi-site clinical research aimed at advancing cancer biomarker discovery and personalized medicine.

**Data availability statement:** The WES data used for pipeline comparisons has been deposited at the database of Genotypes and Phenotypes (dbGaP) under the accession number phs002295.v1.p1. The code for the pipelines described in this manuscript can be found: https://github.com/NCI-CIDC/cidc_wes2_releases and https://github.com/NCI-CIDC/cidc_rnaseq2_releases.

**Funding:** Funder Name: National Cancer Institute Grant ID: 140D0421F0589.

**Competing interests:** The authors have declared that no competing interests exist.

**Abbreviations:** CIMAC-CIDC, The Cancer Immune Monitoring and Analysis Centers – Cancer Immunologic Data Center; NCI, the National Cancer Institute; WES, Whole Exome Sequencing; RNA-Seq, RNA sequencing; GCP, Google Cloud Platform; PACT, the Partnership for Accelerating Cancer Therapies; AML, Acute Myelogenous Leukemia; NSCLC, squamous non–small cell lung carcinoma; CyTOF, cytometry by time of flight; PBMC, Peripheral Blood Mononuclear Cell; NGS, Next-Generation Sequencing; DFCI, the Dana-Farber Cancer Institute; QC, Quality Control; CNVs, Copy Number Variants; Indels, insertions-deletions; dbGaP, the database of Genotypes and Phenotypes; RPKM, per kilobase per million reads.

## Introduction

The Cancer Immune Monitoring and Analysis Centers – Cancer Immunologic Data Center (CIMAC-CIDC) Network, established in 2017 under the Cancer Moonshot Initiative and the Partnership for Accelerating Cancer Therapies (PACT), supports biomarker identification and correlation with clinical outcomes across immuno-oncology trials [1]. Comprising four academic centers (CIMACs) with expertise in cancer immunotherapy, the network provides validated, harmonized immune profiling assays. The CIDC serves as a data coordinating center, offering computational resources for biomarker data storage, distribution, analysis, and integration with clinical data. The network's comprehensive profiling of specimens using standardized assays has demonstrated the success in identifying biomarkers of response, resistance, and toxicity in immunotherapy trials for acute myelogenous leukemia (AML), squamous non–small cell lung carcinoma (NSCLC), and Hodgkin lymphoma [2–4]. The AML study highlighted the potential of immune profiling to guide personalized therapy by identifying biomarkers for patient stratification and novel therapeutic targets using techniques like RNA sequencing (RNA-seq), whole exome sequencing (WES), single cell RNA and T-cell receptor sequencing [2]. In the NSCLC study, the CIMAC-CIDC platform employed techniques like WES, multiplex immunofluorescence, Olink, and NanoString gene expression profiling to identify immune signatures predictive of survival, providing insights into the tumor microenvironment and aiding the development of predictive models for tailored immunotherapy [3]. The combined analysis of mass cytometry (or cytometry by time of flight (CyTOF)), Olink, serology, and T-cell receptor sequencing contributed to improved response rates and overall survival benefits in a trial for combination checkpoint therapy of Hodgkin lymphoma, underscoring the importance of multi-omic and integrative data analyses in improving patient outcomes across different cancers [4]. These findings underscore the importance of multi-omic and integrative data analyses in improving patient outcomes across different cancers.

Data from different laboratory sites often suffer from technical variations [5–7]; standardizing quality control measures and harmonizing protocols are essential to ensure consistent methods and data collection, enabling accurate comparisons and unified analyses across different studies and sites. The CIMAC-CIDC network selected biomarker modules for clinical trials and successfully applied assay validation and harmonization for assays assessing genomics (WES), transcriptomics (RNA-seq), and phenotypic characterization of tumor (mIHC/mIF) and peripheral blood mononuclear cell (PBMC) subtypes (CyTOF), leading to a a high level of comparability between participating laboratories [1,8–10]. The CIDC has developed data standards and software for recording molecular, clinical, and metadata generated by the network, and it hosts centralized pipelines for standardized data processing, acting as the central bioinformatics platform and database for biomarker and clinical data from trials in conjunction with the CIMACs [1].

Updating and benchmarking Next-Generation Sequencing (NGS) pipelines is crucial for maintaining the accuracy, efficiency, and reliability of genomic data analysis. As technology and computational tools evolve, regularly updating these pipelines ensures they incorporate the latest algorithms, reference datasets, and best

practices, which can significantly improve the quality of results. Benchmarking is equally important, as it allows for the validation of pipeline performance against established standards, ensuring that they deliver consistent and reproducible outcomes. This is particularly vital in clinical and research settings, where precise data interpretation can directly impact scientific discoveries and patient care.

The CIDC was originally hosted and maintained by the Dana-Farber Cancer Institute (DFCI), but was migrated to the National Cancer Institute (NCI) in July 2023. The pipelines for WES, RNA-seq, and ATAC-seq were developed over five years ago and have since played a critical role in advancing cancer research. The CIDC's WES and RNA-seq pipelines were comprehensively described in a harmonization study that outlined strategies and evaluated data from HapMap cell lines and NSCLC tumor samples. This study, conducted across diverse platforms and tissue preparation methods, established standardized protocols for WES and RNA-seq library preparation, as well as quality control (QC) metrics [8], ensuring data reliability and reproducibility across studies. As part of the CIDC migration from DFCI to NCI, it became essential to rebuild three pipelines—WES, RNA-seq, and ATAC-seq—using updated software versions and incorporating new modules. These pipelines were restructured to operate on NCI-controlled cloud resources and the Google Cloud Platform (GCP) environment, ensuring improved performance, scalability, and compliance with evolving bioinformatics standards.

In this study, we outline the process of reviewing and updating bioinformatics pipelines, striking a balance between continuity and improvement. We present a streamlined approach for developing pipelines using open-source software, designed for efficient deployment on cloud platforms via Docker containers. This capability leverages the scalability and flexibility of cloud computing, facilitating enhanced data sharing and reproducibility. Benchmarking results demonstrate performance improvements, and a case study highlights the impact of these advancements on clinical research outcomes.

## Materials and methods

### Data selection and processing

Truth set data were selected for Whole Exome Sequencing and RNA-Seq based on public availability of well-documented sequencing and analysis protocols as well as input data (e.g., fastq or BAM files) and results. For WES, The source fastq files for small variant calling were retrieved from NIST's Genome in a Bottle (GIAB) project, a consortium dedicated to authoritative characterization of the human genome. As such, they distribute high-quality sequencing data and benchmarking datasets, which were used to evaluate pipeline performance for small variant calling (S2 Table). For CNV benchmarking, data from the triple negative breast cancer line HCC1395 that has been extensively characterized will be used (S2 Table). These data have previously been used in a comprehensive CNV caller benchmarking study [11].

To evaluate the quantification accuracy of the RNA-Seq pipeline, we used datasets available from ENCODE. These samples were selected from ENCODE's deeply profiled cell lines and are derived from GM12878 and K562, the Tier I cell lines, which are the mostly widely distributed and analyzed cell lines. All analyzed datasets have been processed through their uniform processing pipelines using defined pipelines and parameters. The data were selected based on availability of paired-end total RNA-Seq data (S3 Table). An additional standard dataset that has been described previously [12] was also used to evaluate RNA-Seq quantification performance. This dataset includes 9 replicates samples of hepatocellular cell line MHCC97H with paired-end 2x150 paired end sequencing data. The expression data is presented as reads per kilobase per million reads (RPKM). To measure the accuracy of the fusion calls, simulated data was be used. The reads are generated by the Broad for fusion caller benchmarking and are generated as 101 bp paired-end reads (S3 Table). The data have been previously used to benchmark various fusion callers. As these are simulated reads, the truth set consisted of the known fusions of the input data.

### Bioinformatic pipelines

WES/RNAseq truth set data were processed through the original pipelines as described previously [9]. The enhanced pipeline reference files and software versions are described in S4 Table and S1 and S2 Figs, respectively. For the

purposes of validation, the Snakemake pipelines were run at the command line with standard invocation of the snake-make executable (e.g., snakemake –cores 60). The pipelines allow user configurable modifications prior to pipeline runs. These configurations are outlined in the pipelines respective config.yaml files (WES: https://github.com/NCI-CIDC/cidc_wes2_releases/blob/main/config/config.yaml; RNA-Seq: https://github.com/NCI-CIDC/cidc_rnaseq2_releases/blob/main/config/config.yaml). Configurable parameters include such options as output directories, cores used for the entire pipeline run and specific rules and read trimming length. However, it is noted that these parameters are left unchanged for all pro-ductions analyses and for all validation analyses described in this study, in keeping with the prevailing CIMAC-CIDC goal of maintaining consistent, standardized data analysis. The pipelines were deployed on virtual machines on the google cloud platform (GCP) running Ubuntu 20.04.6 LTS. After validation and for the purposes of analyzing production data, the pipelines were containerized with Docker to ease deployment and minimize analytical variability.

### Data and statistical analysis

VCF Comparisons were performed using hap.py (https://github.com/Illumina/hap.py) as recommended by GIAB for analysis. CNV comparisons were performed using an in-house comparison script, with matching CNV regions between datasets being those that overlap by>= 90%. RNA-Seq quantification spearman correlations were performed on log-transformed TPM data using R statistical software. Fusion analysis was performed by comparing pipeline output data to the known fusion calls to identify true positives (TP), false positives (FP) and false negatives (FN). Recall was calculated as TP/ TP+FN and precision was calculated as TP/ TP+FP. The Jaccard index for evaluation of fusion reproducibility was calculated as

$$J(X,Y) \ = \ |X \cap Y| \ / \ |X \cup Y|$$

using an in-house script. The OncoKB Cancer Gene List was retrieved from https://www.oncokb.org/cancer-genes on August 20th 2024.

## Results

The Cancer Immunologic Data Center (CIDC) is responsible for ingesting, distributing, and sharing clinical and assay data generated by the CIMAC-CIDC immuno-oncology clinical research network. For certain assays, the CIDC also performs data analysis using standardized bioinformatics pipelines. Previously, the CIDC's RNA sequencing (RNA-Seq) and whole exome sequencing (WES) pipelines have been described [8]. The combination of these standardized pipelines with har-monized assays offers comprehensive immune profiling across a wide range of cancer types, aiming to identify biomark-ers of response, resistance, and toxicity.

As standard assays and analysis techniques in the field evolve, the CIDC pipelines must evolve as well. However, in line with the CIMAC-CIDC network's goal to produce standardized, comparable results across clinical trials and time points, pipeline upgrades must be thoroughly documented, maintain existing functionality, allow for consistency of results pre- and post-pipeline upgrades, and when changes are introduced, the changes must provide clear benefits. Before updating the pipelines, the CIDC bioinformatics development team outlined five primary goals (Table 1) to guide the design: improving code readability and consistency for future maintenance development, addressing security vulnerabil-ities or software end-of-life issues, modernizing the software to meet current industry standards, enhancing functionality as required by the network, and increasing transparency and accessibility. The last goal is in keeping with the Cancer Moonshot Initiative's over-arching goal of accelerating discovery in cancer research, and includes the move from licensed, proprietary software to open-source software and public releases of the CIDC's bioinformatic pipelines for use by the broader oncology research community.

With these goals in mind, enhanced pipeline designs were created, refined, and approved before implementation. Snakemake was chosen for the underlying workflow management system largely because the original pipelines also

**Table 1. Goals and objectives for pipeline enhancements.**

| Category | Goals | Objectives |
|---|---|---|
| **Refactoring** | Restructuring pipeline code to improve readability, thus, making it easier to maintain, debug, and update. | Provide code review to determine areas for refactoring |
| **Maintenance** | Maintain current software versioning to optimize vendor support, application performance and address security flaws. | Provide gap analysis on current software versions to determine what tools to upgrade |
| **Modernization** | Maintain industry standard software to optimize the best combination of biochemistry, mathematics, computer science, data science, and modern data analytics tools. | Review the performance of current packages to determine if current functionality is meeting community needs (CIMAC Input) |
| **Enhancements** | Provide enhancements to pipeline to improve current functionality/performance, while maintaining backward compatibility with previous versions. | Add new functionality and features as desired by the stakeholder community |
| **Accessibility** | Updating the pipeline to allow it to be a valuable resource to the broader oncology research community | Where possible, move from licensed to open-source software and create public repository releases |

were developed with Snakemake. Other workflow management systems, such as Nextflow, were evaluated as well. While these other tools have excellent documentation, community support, and in many cases modifiable, publicly available pipelines, the desire to retain consistency and traceability between the original and enhanced pipelines led to the selection of Snakemake. To improve design efficiency, a Snakemake-based pipeline template was developed (Fig 1). The newly created Snakemake template standardizes code and data structures for input files, reference files, output files, and analysis rules across all pipelines hosted by the CIDC. It takes advantage of Snakemake's workflow management tools, such as automated logging, parallelization, checkpointing, and environment management through Conda integration. A custom set of tools was also developed to facilitate data file movement between Snakemake workflows and the existing Google Cloud Storage ecosystem. Given that the workflows primarily process short-read sequencing data, built-in quality control tests focused on this datatype have been added to the template.

To further streamline deployment, the pipelines have been packaged as Docker containers, which can be launched on the CIDC's Google Cloud Platform (GCP) virtual machines. This combination of easily deployable containers and cloud resources enables scalable analyses, reducing both runtimes and overall analysis costs.

Using the Snakemake template, schematics of the pipeline designs provide the general analysis workflow and highlight specific changes that have been made to satisfy the aforementioned goals and objectives (S1 and S2 Figs). In this report, we focus on the enhanced RNA-Seq and WES pipelines, as the original pipelines have been previously described and serve as a point of comparison [1]. After pipeline development, the enhanced pipelines were evaluated using rigorous analytical validations designed to evaluate the core functionality of the pipelines, with a focus on reproducibility, precision and recall (S3 Fig). While superior performance in all categories is desirable, performance specifications can often be tuned through altered analysis parameters to favor one metric over another. In keeping with the key objective of the CIMAC-CIDC network in providing standardized results across time and clinical trials, reproducibility and precision were the primary metrics to evaluate and optimize.

For the WES pipeline, the analytical validation focused on the pipeline performance in calling small variants (single nucleotide variants (SNVs) and small (<50 bp) insertions-deletions (indels)) as well as copy number variants (CNVs). Two truth sets were used for these purposes. Small variants were evaluated with datasets and gold-standard results from NIST's Genome in a Bottle consortium (HG001, HG002, HG003, and HG004) [13], while CNVs were evaluated with data and high-confidence call sets from a reference cancer cell line (HCC1395) (S2 Table). Small variant results were analyzed with Illumina's hap.py VCF comparison tool, which allows for standardized comparison of complex variants using imputed haplotype sequences (https://github.com/Illumina/hap.py). As described in the validation schematic (S3 Fig), the pipeline was run in triplicate, with each run performed on a different virtual machine and set up by a different user to measure 'real

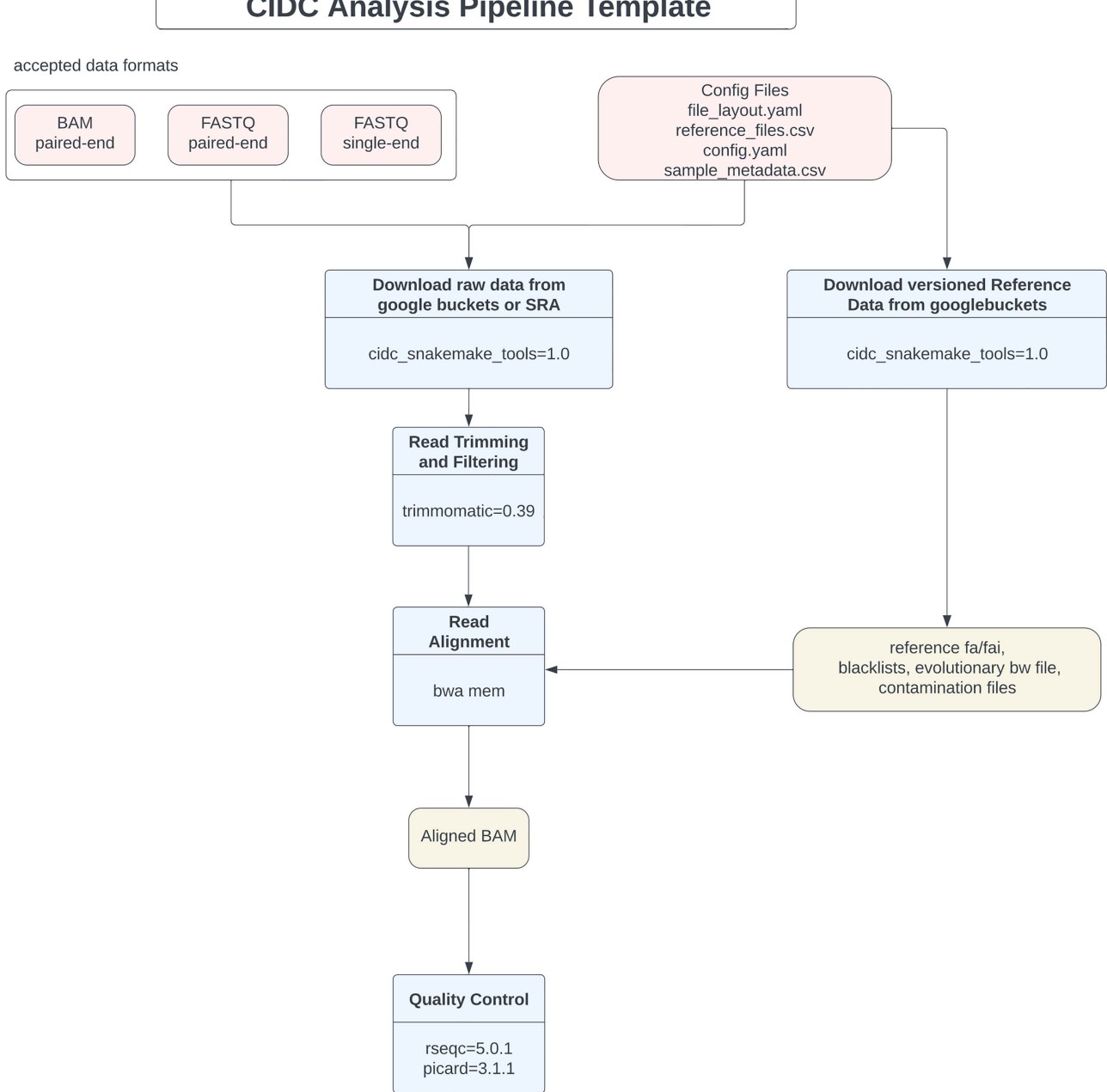

**Fig 1. Snakemake workflow template for enhanced WES and RNA-Seq pipelines.** Diagram of the Snakemake workflow template used for pipeline standardization. The modular structure enables scalable, reproducible workflows with automated logging, parallelization, and environment management through Conda integration, facilitating rapid updates and maintenance.

world' reproducibility. From the validation results, the pipeline reproducibility for the small variants is nearly perfect (Fig 2). Likewise, the precision and recall for calling small variants was high for all samples, with the average recall and precision of ~0.96. Performance was quantitatively better for SNVs as compared to small indels, as expected for most short-read aligners. Pipeline performance was evaluated for CNVs as well (Fig 2). The reproducibility and precision were excellent. A

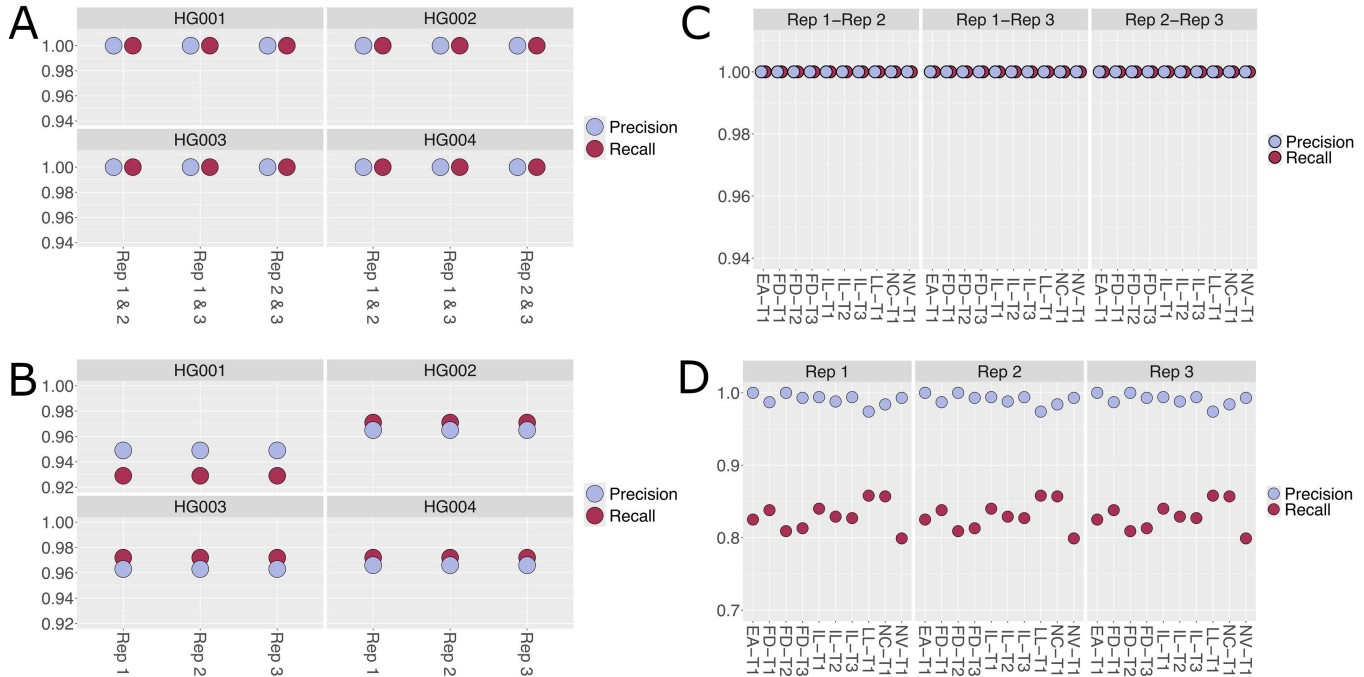

**Fig 2. Validation of enhanced WES pipeline for variant calling and CNV detection.** Analytical validation results for the WES pipeline, focusing on reproducibility, precision, and recall for small variants and copy number variants. Results are shown from triplicate runs, demonstrating nearly perfect reproducibility and high-confidence detection for clinically relevant variants. Rep1: Replicate 1, Rep2: Replicate 2, Rep3: Replicate 3.

reduction in recall was observed as well, though, this effect is keeping in line with the overall design strategy. The pipeline employs three independent CNV callers to analyze the data and reports only the CNVs identified by all three callers. CNV callers are known to give disparate results, particularly when using short-read sequencing data [11] which can negatively affect pipeline recall. In addition, the truth set is based on whole genome sequencing (WGS) data, while the pipeline analyzed WES data. WGS has been shown to be more sensitive for CNV calling, which could also contribute to the observed pipeline recall. Overall, the WES pipeline showed exceptionally reproducible results and produces high confidence results for small variants and CNVs.

As with the WES validation, RNA-Seq pipeline validation was performed in triplicate using two well-established truth sets (S3 Fig). Transcript quantification results were evaluated with a fourteen-sample truth set from the ENCODE project. The ENCODE data were paired-end sequencing data collected from two cell lines (GM12878 and K562) at various timepoints and processed through the ENCODE bulk RNA-Seq quantification pipeline (https://www.encodeproject.org/pipelines/) (S3 Table). Gene fusion calls were evaluated with a truth set released by the STAR-Fusion development team (https://github.com/STAR-Fusion/STAR-Fusion_benchmarking_data). This truth set included five datasets of simulated paired-end 101-bp reads, which were generated to include chimeric transcripts of hundreds of known fusions per set (S3 Table). Transcript quantification data were log transformed prior to comparison to reduce the effect of outliers on the correlation between the experimental and truth set data. As with the WES pipeline, reproducibility was nearly perfect, with for example, transcript quantification correlations of $r^2 > 0.99$ (Fig 3). For evaluating transcript quantification accuracy, the pipeline output was compared against the truth sets. The acceptance criterion for successful correlation metric was set as $r^2 > 0.8$, as this generally represents good agreement between datasets, while values of $> 0.9$ represent excellent agreement. When evaluating the datasets, the average correlation between truth set and pipeline results was $r^2 \sim 0.89$ and thus can be considered good-to-excellent agreement (Fig 3). It was expected that there would not be perfect agreement as the

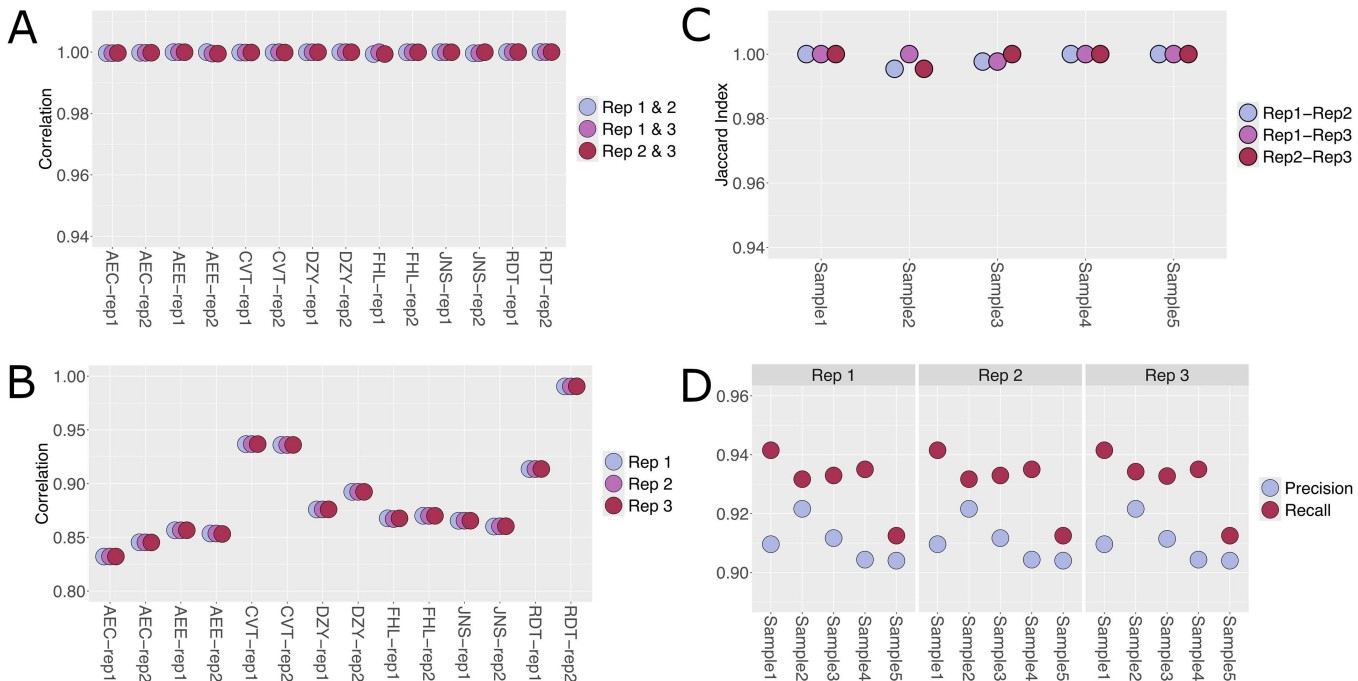

**Fig 3. Validation of enhanced RNA-Seq pipeline for transcript quantification and fusion calling.** Analytical validation results for the RNA-Seq pipeline, focusing on reproducibility, precision, and recall for transcript quantification and fusion callings. Results are shown from triplicate runs, demonstrating nearly perfect reproducibility and high-confidence quantification and fusion detection. Rep1: Replicate 1, Rep2: Replicate 2, Rep3: Replicate 3.

analysis techniques and software differed between the experimental and truth sets. However, these data suggest that the pipeline produces acceptable transcript quantification results that are consistent with widely used standards. As with the transcript quantification results, the enhanced pipeline generated highly reproducible fusion calls. Indeed, only a single discordant call was observed among all replicate runs. Fusion calling precision and recall were quantified using a previously described truth set, and the average value for both metrics was > 0.9, suggesting excellent analytical performance (Fig 3). Thus, the RNA-Seq pipeline met all acceptance criteria and produces high-confidence results.

With the independent truth sets used during pipeline analytical validation, we could objectively show that the enhanced pipelines met our desired performance specifications and produced results comparable to industry standards. However, for this work in particular, it is also important to compare the enhanced pipelines' results to the previously deployed pipelines. This comparison is meant to primarily evaluate continuity of analysis and secondarily, to identify potential benefits associated with the pipeline enhancements. When comparing WES results from the two pipelines, the precision and recall are nearly identical between either pipeline version (Fig 4). Indeed, this result was expected as the primary changes to the WES pipeline were to address software end-of-life or security concerns, and to replace commercially licensed software tools with functionally equivalent open-source tools. Thus, the enhanced WES pipeline provides a comparable level of analytic performance as the previous pipeline without the additional costs associated with software licenses. The RNA-Seq pipeline, in contrast, was enhanced to address known performance issues associated with previous software versions, specifically in terms of alignment robustness and fusion calling accuracy (S2 Fig). The comparison of the two RNA-Seq pipeline versions showed comparable results for transcript quantification for the majority of samples (Fig 4 and S4 Fig). However, the original pipeline demonstrated markedly poor correlation metrics for two samples with very low read depths. The enhanced pipeline, in contrast, generated transcript quantification results that were highly correlated

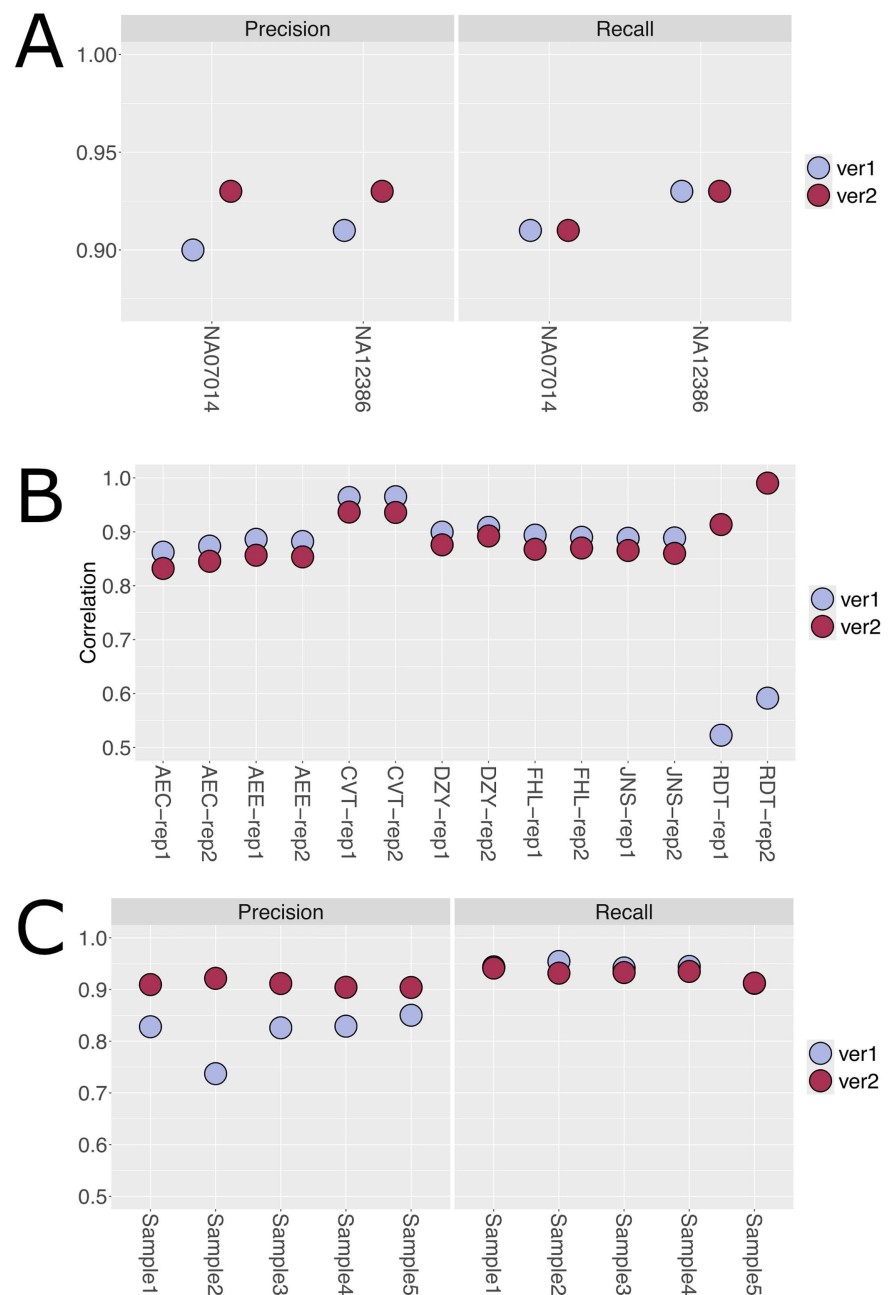

**Fig 4. Comparison of the analytical performance of the extant and enhanced pipelines. A.** Comparison of small variant precision and recall for the two pipeline versions. As shown the move to open-source software did not impact enhanced pipeline performance. **B & C**. Comparative analysis of transcript quantification (correlation) and fusion calling metrics (precision and recall) between the original and enhanced RNA-Seq pipelines. The enhanced pipeline shows improved fusion detection precision and consistency across samples, particularly for low-read-depth samples. Ver1: Version 1 (Original) Pipeline, Ver2: Version 2 (Enhanced) Pipeline.

to the truth set. The fusion calling metrics, specifically in terms of precision, were markedly improved by the enhanced pipeline (Fig 4). The core fusion calling software STAR-Fusion was upgraded from version 1.7 to 1.12. Earlier versions of STAR-Fusion were known to produce a number of false positive calls, and the upgraded software addresses this issue

(https://github.com/STAR-Fusion/STAR-Fusion/releases/tag/STAR-Fusion-v1.3.1). Thus, the incorporation of the upgraded software generates predictable performance improvements for the enhanced pipeline.

The most significant improvement between the original and enhanced RNA-Seq pipelines was observed in fusion call accuracy, particularly in reducing false positives. To evaluate this in detail, we analyzed false positives uniquely identified by the original pipeline that were absent in the enhanced pipeline's results. For convenience, we label these false positives as $FP_{original-only}$ and these represent fusions erroneously called by the original pipeline, but due to software improvements, are correctly filtered out by the enhanced pipeline. On average, the original pipeline produced approximately 64 $FP_{original-only}$ calls per sample, which were effectively filtered out by the enhanced pipeline (Fig 5).

To determine the potential biological relevance of these false positive calls, we cross-referenced the gene partners from the $FP_{original-only}$ list with OncoKB's Cancer Gene List, a comprehensive database of genes frequently implicated in cancer. This analysis revealed that 20 of the gene partners were flagged as cancer-related by at least one source in OncoKB (Fig 5). Upon further refinement—requiring validation by three or more sources—six unique genes were identified in the $FP_{original-only}$ list, including BCOR, a known tumor suppressor associated with oncogenic fusions [14].

These findings demonstrate that the original pipeline occasionally identified false positives in plausible cancer-related fusions. The enhanced pipeline which implements more recent software releases reduces these potentially mis-leading

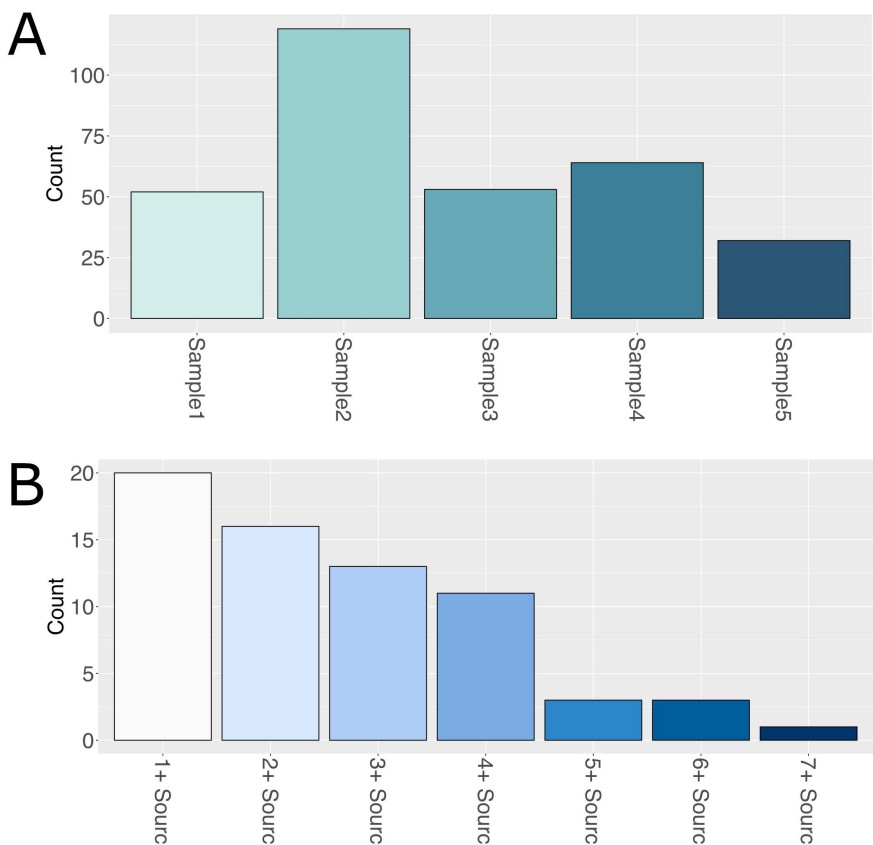

**Fig 5. Evaluation of false positive fusion calls and relevance to OncoKB cancer gene list. A.** Assessment of false-positive fusion calls identified uniquely by the original RNA-Seq pipeline but correctly filtered out by the enhanced pipelines (i.e., $FP_{original-only}$). **B.** Gene partners from **A.** were cross-referenced with the OncoKB Cancer Gene List to evaluate potential clinical relevance. The number of sources is those within OncoKB that support the gene partner as potentially oncogenic.

false positive calls, resulting in more accurate and high-confidence fusion detection. By eliminating biologically irrelevant noise, the enhanced pipeline supports clinical research with improved reliability and clarity. Overall, this comparison highlights the enhanced pipeline's ability to maintain analytical continuity while offering improvements in specificity, which is crucial for the accurate interpretation of biological data.

## Discussion

Here we present a generalized compartmentalized workflow framework using Snakemake and Docker for the high-throughput and standardized analysis of large scale NGS datasets. With the demands for standardization of NGS analysis within the CIMAC-CIDC consortium, we sought to devote the resources for this undertaking to minimize effort in reanalysis going forward. We demonstrate that the approach we utilized here is scalable, distributable, and reproducible. It also presents a roadmap for future updates both within the CIMAC-CIDC network and other consortiums dedicated to the production and analysis of standardized datasets.

This study highlights the critical role of maintaining and enhancing bioinformatics pipelines to support evolving analytical standards within the CIMAC-CIDC network. By integrating updated software versions, open-source toolkits, and workflow management, we address both functional and operational needs essential for handling the scale and complexity of next-generation sequencing (NGS) data in a multi-institution clinical trial setting. Notably, the decision to shift from commercially licensed software to open-source alternatives has ensured sustainability and cost-efficiency, facilitating broader accessibility without sacrificing analytical rigor. This shift also aligns with the network's goal of creating reproducible, scalable workflows capable of delivering standardized data across studies and time points, which is fundamental for multi-site collaborations where consistency in data analysis and interpretation is paramount.

A key advantage of the Snakemake-based pipeline template is its modular design, which allows individual components to be updated or replaced without disrupting the overall workflow. This modularity facilitates efficient maintenance, as specific pipeline modules can be modified or upgraded to incorporate new software versions, enhanced quality control checks, or emerging computational techniques and algorithms. As a result, the pipelines remain agile, adapting quickly to evolving analytical requirements within the CIMAC-CIDC network while retaining a stable and consistent framework.

Moreover, the extensible nature of this template ensures that additional analysis pipelines can be seamlessly developed as new assays, such as single-cell RNA sequencing or spatial transcriptomics, are integrated into the network's capabilities. By leveraging the same standardized Snakemake framework, new assays can be implemented with minimal reconfiguration, preserving the continuity of pipeline management and data standards. This template thus provides a scalable foundation for the CIMAC-CIDC network, equipping it to incorporate and support future bioinformatics needs as clinical and research demands grow.

To meet the specific clinical data requirements of the CIMAC-CIDC network, we carefully optimized the pipelines, and if necessary, opted to prioritize specificity over sensitivity. This decision reflects a deliberate trade-off aimed at ensuring that biomarkers identified are highly correlated with clinically relevant phenotypes, minimizing the risk of false positives that could complicate interpretation in a clinical context. By emphasizing specificity, we enhance confidence that detected biomarkers are both robust and biologically meaningful, which is especially critical in studies where these findings may influence subsequent trial design, therapeutic decisions or patient stratification. Additionally, reproducibility was maintained as the foremost criterion, given the necessity for reliable results across diverse settings, including multi-laboratory environments, various clinical trials, and extended timespans. This stringent reproducibility requirement ensures that any biomarker or genetic feature detected is not influenced by pipeline variability, thereby supporting the network's goal of generating dependable, longitudinal data suitable for high-impact clinical research and analysis.

Our benchmarking and validation efforts demonstrate that the new pipelines' specifications match or exceed that of the original pipelines, particularly for RNA-seq fusion detection—a key area for biomarker discovery in immuno-oncology. The cloud-deployment framework further enhances the pipeline's utility by providing flexible scaling and optimized resource

use, supporting the consortium's large-scale data requirements without the constraints of local computing infrastructure. This cloud-based approach allows for consistent application of pipeline updates, version control, and real-time improvements across the CIDC, ensuring the alignment of bioinformatics workflows with the most current analytical standards and methods. The approach is also flexible and cost-efficient, such that resources are appropriately distributed across the entire network.

However, cloud computing also presents challenges, particularly concerning data security, management, and cost. While our framework is built within the latest security standards offered by NCI's cloud services, ongoing monitoring and threat detection are necessary as technology and security protocols evolve. Furthermore, while the enhanced pipelines have shown excellent precision and recall for variants, certain biological interpretations, particularly for low-frequency or complex structural variants, may still require further optimization or future updates. The framework described in this work can be used to help develop and deploy such updates.

## Conclusions

The enhanced pipelines presented here offer a replicable model for modern bioinformatics in clinical research. By balancing rigorous analytical validation with flexible, scalable infrastructure, this work not only advances the CIMAC-CIDC network's capabilities but also sets a precedent for future consortiums aiming to support high-throughput, standardized multi-omic analyses across diverse research and clinical settings.

## Supporting information

**S1 Fig. Schematic of CIDC WES pipeline.** (left) Specific pipeline changes associated with the enhanced pipeline and the rationale for the changes are listed. (right) Workflow with the key software and software versions associated with each step shown.
(TIF)

**S2 Fig. Schematic of CIDC RNA-Seq pipeline.** (left) Specific pipeline changes associated with the enhanced pipeline and the rationale for the changes are listed. (right) Workflow with the key software and software versions associated with each step shown.
(TIF)

**S3 Fig. Schematic of the analytical validation workflow.** All Validation runs were experimental replicates, with a unique user and virtual machine (VM) being used for each run. In total, the validation was performed in triplicate to determine intra-pipeline reproducibility. The output of the triplicate runs was also compared pre-selected truth sets to measure precision and recall, along with other metrics.
(TIF)

**S4 Fig. Comparison of the analytical performance of the original and enhanced RNA-Seq pipelines.** Comparative analysis of transcript quantification metrics between the original and enhanced RNA-Seq pipelines. The supplemental analysis includes evaluation of samples described in Lu S, et al. [12].
(TIF)

**S1 Table. False positive gene partners with 3 or more sources on OncoKB's Cancer Gene List uniquely identified in the fusion calls of the original pipeline (i.e., FP$_{original-only}$).**
(XLSX)

**S2 Table. Description and source of data and truth sets used in evaluation of the WES pipeline.**
(XLSX)

**S3 Table. Description and source of data and truth sets used in evaluation of the RNA-Seq pipeline.**
(XLSX)

**S4 Table. Reference file information for CIDC enhanced WES and RNA-Seq pipelines.**
(XLSX)

## Acknowledgments

Scientific support was supplied by the National Cancer Institute (NCI) and the Partnership For Accelerating Cancer Therapies (PACT) consortium. We would like to acknowledge the support and helpful discussions provided by the CIMAC-CIDC network researchers, CIMAC-CIDC Project Managers and NCI Project Managers, including David Patton, Fatma Onmus, Magdalena Thurin, Rebecca Enos, Rebecca Venediktov, Sacha Gnjatic, and Edgar Gonzalez-Kozlova.

## Author contributions

**Conceptualization:** Chunhua Yan, Qingrong Chen, Nicholas Renzette, Umit Topaloglu.

**Data curation:** Cu Nguyen, Trinh Nguyen, Gloria Trivitt, Brian Capaldo, Nicholas Renzette.

**Formal analysis:** Cu Nguyen, Gloria Trivitt, Brian Capaldo, Nicholas Renzette.

**Funding acquisition:** Gloria Trivitt, Nicholas Renzette.

**Methodology:** Brian Capaldo, Chunhua Yan, Qingrong Chen, Nicholas Renzette, Umit Topaloglu, Daoud Meerzaman.

**Software:** Cu Nguyen, Trinh Nguyen, Gloria Trivitt, Brian Capaldo, Nicholas Renzette.

**Supervision:** Chunhua Yan, Qingrong Chen, Daoud Meerzaman.

**Validation:** Cu Nguyen, Trinh Nguyen, Gloria Trivitt, Brian Capaldo, Nicholas Renzette.

**Visualization:** Cu Nguyen, Trinh Nguyen, Gloria Trivitt, Nicholas Renzette.

**Writing – original draft:** Nicholas Renzette.

**Writing – review & editing:** Cu Nguyen, Trinh Nguyen, Gloria Trivitt, Brian Capaldo, Chunhua Yan, Qingrong Chen, Umit Topaloglu, Daoud Meerzaman.

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
