## [Decision Letter · Decision Letter 0]

11 Jun 2025

PONE-D-25-08135Modular and Cloud-Based Bioinformatics Pipelines for High-Confidence Biomarker Detection in Cancer Immunotherapy Clinical TrialsPLOS ONE

Dear Dr. Renzette,

Thank you for submitting your manuscript to PLOS ONE. After careful consideration, we feel that it has merit but does not fully meet PLOS ONE’s publication criteria as it currently stands. Therefore, we invite you to submit a revised version of the manuscript that addresses the points raised during the review process.

We look forward to receiving your revised manuscript.

Kind regards,

Surya Saha, PhD

Academic Editor

PLOS ONE

Journal Requirements:

[Funder Name: National Cancer Institute

Grant ID: 140D0421F0589]. 

3. Thank you for stating the following in your manuscript:

[Additional support is made possible through the NCI Clinical Reporting and Research Informatics (CRRI) Support Services Contract 140D0421F0589 (Essex Management)]

[Funder Name: National Cancer Institute

Grant ID: 140D0421F0589]. 

4. We notice that your supplementary figures and tables are included in the manuscript file. Please remove them and upload them with the file type 'Supporting Information'. Please ensure that each Supporting Information file has a legend listed in the manuscript after the references list.

Reviewers' comments:

Reviewer's Responses to Questions

**Comments to the Author**

1. Is the manuscript technically sound, and do the data support the conclusions?

Reviewer #1: Yes

Reviewer #2: Yes

2. Has the statistical analysis been performed appropriately and rigorously? 

Reviewer #1: Yes

Reviewer #2: Yes

3. Have the authors made all data underlying the findings in their manuscript fully available?

Reviewer #1: Yes

Reviewer #2: Yes

4. Is the manuscript presented in an intelligible fashion and written in standard English?

Reviewer #1: Yes

Reviewer #2: Yes

5. Review Comments to the Author

Reviewer #1: It would be useful to see the list of gene partners from the false positive list that overlap with OncoKB's Cancer gene list as a supplementary table, or provide a list of true positives as a supplementary table. If these were indeed biologically irrelevant noise, then why point out that BCOR is associated with oncogenic fusions -- isn't this an argument against using the enhanced pipeline? Perhaps focusing on a gene on the true positive list that is associated with an oncogenic signal would help to recapitulate the need for the enhanced pipeline that gives more accurate fusion calls.

Reviewer #2: The paper describes the establishment of updated open-source bioinformatics workflows for WES and RNA-seq data by the Cancer Immune Monitoring and Analysis Centers - Cancer Immunologic Data Center (CIMAC-CIDC). Previous existing pipelines were updated, benchmarked and results compared also with those of previous pipelines.

The paper provides a useful example of pipelines validation process, highlighting ground truth datasets to be utilized for this purpose as well as benchmarking measures to assess also reproducibility of pipelines. Yet the paper should and could be improved by:

1. giving more information on how the pipelines have been built and why, e.g., it was decided to favor Snakemake vs. e.g. nf-core community based pipelines;

2. giving more information on how pipelines can be used and what choices are user dependent. It is not very clear how reprodicibility between users is assessed if it is not clear what choices are left to users;

3. improving presentation by carefully revising language (in some points of the paper entire sentences are repeated, e.g, lines 48-51, and there are some other errors) and improving figures' quality (Figures 2-4 are too small and not well readable).

6. PLOS authors have the option to publish the peer review history of their article (what does this mean? ). If published, this will include your full peer review and any attached files.

**Do you want your identity to be public for this peer review?** For information about this choice, including consent withdrawal, please see our Privacy Policy .

Reviewer #1: No

Reviewer #2: No

---

## [Author Response · Author response to Decision Letter 1]

22 Jul 2025

From the Academic Editor

Response: We have reformatted the manuscript, tables and figures to match PLOS ONE’s style requirements.

[Funder Name: National Cancer Institute

Grant ID: 140D0421F0589].

Response: The funders had no role in study design, data collection and analysis, decision to publish, or preparation of the manuscript. Our cover letter has been updated with this information and we appreciate you updating the submission on our behalf.

3. Thank you for stating the following in your manuscript:

[Additional support is made possible through the NCI Clinical Reporting and Research Informatics (CRRI) Support Services Contract 140D0421F0589 (Essex Management)]

[Funder Name: National Cancer Institute

Grant ID: 140D0421F0589].

Response: We have removed funding information from the Acknowledgements section. Our funding statement should read:

Scientific and financial support for the Cancer Immune Monitoring and Analysis Centers-Cancer Immunologic Data Center (CIMAC-CIDC) Network are provided through the National Cancer Institute (NCI) through NCI contract 140D0421D0007 to the CIDC operated by NCI. Additional support is made possible through the NCI Clinical Reporting and Research Informatics (CRRI) Support Services Contract 140D0421F0589 (Essex Management).

Additional financial support for the CIMAC-CIDC Network is provided by the Partnership for Accelerating Cancer Therapies (PACT) public-private partnership (PPP) made possible through funding provided to the Foundation for the National Institutes of Health (FNIH) by: AbbVie Inc., Amgen Inc., Boehringer-Ingelheim Pharma GmbH & Co. KG., Bristol-Myers Squibb, Celgene Corporation, Genentech Inc, Gilead, GlaxoSmithKline plc, Janssen Pharmaceutical Companies of Johnson & Johnson, Novartis Institutes for Biomedical Research, Pfizer Inc., and Sanofi.

Our cover letter has been updated this information.

4. We notice that your supplementary figures and tables are included in the manuscript file. Please remove them and upload them with the file type 'Supporting Information'. Please ensure that each Supporting Information file has a legend listed in the manuscript after the references list.

Response: We have removed the supplementary figures and tables and saved them as separate ‘Supporting Information’ files. The associated legends are included in the manuscript after the reference list.

Response: The list has been reviewed and confirmed to be accurate and devoid of retracted publications.

From the Reviewers:

Reviewer #1: It would be useful to see the list of gene partners from the false positive list that overlap with OncoKB's Cancer gene list as a supplementary table, or provide a list of true positives as a supplementary table. If these were indeed biologically irrelevant noise, then why point out that BCOR is associated with oncogenic fusions -- isn't this an argument against using the enhanced pipeline? Perhaps focusing on a gene on the true positive list that is associated with an oncogenic signal would help to recapitulate the need for the enhanced pipeline that gives more accurate fusion calls.

Response: We believe the reviewer makes an excellent point with their comment and highlights a misunderstanding about the presented data. The false positives discussed in the manuscript are those that are uniquely outputted from the original pipeline but filtered out by the enhanced pipelines described in the manuscript. In other words, it represents the significant portion of fusions that were erroneously called by the original pipeline but are no longer produced by the enhanced pipelines, thus showing the utility of the enhanced pipeline. We then investigated these false positives further by comparing to the OncoKB Cancer Gene List, and identify a subset of fusion calls with biological relevance. Thus, the original pipeline produced ‘red herring’ false positives that could lead to incorrect biological and clinical interpretations, but since the enhanced pipeline filters these fusions out, it is no longer an issue for the enhanced pipelines.

While this is the intent of the section, it was not clearly stated. We have used this as an opportunity to clean up the language of that section to make this important distinction more accessible to the readers. We specifically label this class of false positives as FPoriginal-only for a convenient shorthand and easier interpretation by the reader.

Reviewer #2: The paper describes the establishment of updated open-source bioinformatics workflows for WES and RNA-seq data by the Cancer Immune Monitoring and Analysis Centers - Cancer Immunologic Data Center (CIMAC-CIDC). Previous existing pipelines were updated, benchmarked and results compared also with those of previous pipelines.

The paper provides a useful example of pipelines validation process, highlighting ground truth datasets to be utilized for this purpose as well as benchmarking measures to assess also reproducibility of pipelines. Yet the paper should and could be improved by:

1. giving more information on how the pipelines have been built and why, e.g., it was decided to favor Snakemake vs. e.g. nf-core community based pipelines;

Response: We thank the reviewer for this comment, as it was one that we considered during pipeline planning and development. Indeed, our team has had considerable experience and success deploying nf-core pipelines in the past. Ultimately, the decision was a practical one. The original pipelines were written with Snakemake and for ease of transition and with the desire for continuity, we chose to remain with Snakemake. The decision in no way represents a technical endorsement or critique of either (or other) workflow management tools. We have updated the text with the following information (Lines 159-164):

Snakemake was chosen as the workflow management system largely because the original pipelines also were developed with Snakemake. Other workflow management systems, such as Nextflow, were evaluated as well. While these other tools have excellent documentation, community support, and in many cases modifiable, publicly available pipelines, the desire to retain consistency and traceability between the original and enhanced pipelines led to the selection of Snakemake.

2. giving more information on how pipelines can be used and what choices are user dependent. It is not very clear how reprodicibility between users is assessed if it is not clear what choices are left to users;

Response: The reviewer comment highlights an important point that was missing from the manuscript but has since been corrected. The pipelines are configurable though as a rule, the configurations do not change. The CIDC is the central processing hub for the CIMAC-CIDC network and a key mandate for the CIDC is to maintain consistent, standardized processing pipelines. As such, the configurations are unchanged for all productions runs and for all runs described in the manuscript. We thank the reviewer for helping us to make this clear to the readers with the following text (Lines 122-130).

The pipelines allow user configurable modifications prior to pipeline runs. These configurations are outlined in the pipelines respective config.yaml files (WES: https://github.com/NCI-CIDC/cidc_wes2_releases/blob/main/config/config.yaml ; RNA-Seq: https://github.com/NCI-CIDC/cidc_rnaseq2_releases/blob/main/config/config.yaml). Cofigurable parameters include such options as output directories, cores used for the entire pipeline run and specific rules, and read trimming length. However, it is noted that these parameters are left unchanged for all productions analyses and for all validation analyses described in this study, in keeping with the prevailing CIMAC-CIDC goal of maintaining consistent, standardized data analysis.

3. improving presentation by carefully revising language (in some points of the paper entire sentences are repeated, e.g, lines 48-51, and there are some other errors) and improving figures' quality (Figures 2-4 are too small and not well readable).

Response: We appreciate the feedback and have used this opportunity to refine the text and figures as highlighted by the reviewer. There are numerous changes throughout the manuscript which can be found in the track changes version included in the re-submission.

We have also re-created all figures in the manuscript to for improved readability and for compliance with PLOS’s figure guidelines.

---

## [Editor Report · Decision Letter 1]

7 Aug 2025

Modular and cloud-based bioinformatics pipelines for high-confidence biomarker detection in cancer immunotherapy clinical trials

PONE-D-25-08135R1

Dear Dr. Renzette,

Congratulations!!

We’re pleased to inform you that your manuscript has been judged scientifically suitable for publication and will be formally accepted for publication once it meets all outstanding technical requirements.

Kind regards,

Surya Saha, PhD

Academic Editor

PLOS ONE
---

## [Editor Report · Acceptance letter]

PONE-D-25-08135R1

PLOS ONE

Dear Dr. Renzette,

I'm pleased to inform you that your manuscript has been deemed suitable for publication in PLOS ONE. Congratulations! Your manuscript is now being handed over to our production team.

Kind regards,

on behalf of

Dr. Surya Saha

Academic Editor

PLOS ONE